# Hydroxychloroquine reduces T cells activation recall antigen responses

**Monika M. Kowatsch**[1]☯, **Julie Lajoie**[1,2]☯, **Lucy Mwangi**[2], **Kenneth Omollo**[2],
**Julius Oyugi**[1,3,4], **Natasha Hollett**[1], **Joshua Kimani**[1,3,4], **Keith R. Fowke**[1,2,3,5]*

1 Laboratory of Viral Immunology, Department of Medical Microbiology and Infectious Diseases, University of Manitoba, Winnipeg, Manitoba, Canada, 2 Department of Medical Microbiology, University of Nairobi, Nairobi, Kenya, 3 Partners for Health and Development in Africa, Nairobi, Kenya, 4 University of Nairobi Institute for Tropical and Infectious Diseases, University of Nairobi, Nairobi, Kenya, 5 Department of Community Health Science, University of Manitoba, Winnipeg, Manitoba, Canada

☯ These authors contributed equally to this work.
* Keith.Fowke@umanitoba.ca

## Abstract

### Background

In the context of the current COVID-19 pandemic, there is still limited information about how people suffering from autoimmune diseases respond to the different COVID vaccines. The fact that they are taking an immunosuppressant or other drugs that aim to decrease the immune system activities, such as hydroxychloroquine (HCQ), could also impact their ability to respond to a COVID vaccine and vaccines in general.

### Methods

Heathy donors were given 200mg of HCQ daily for 6-weeks to assess HCQs impact on the systemic T cells and humoral immune response. Peripheral blood mononuclear cells (PBMC) and plasma were obtained at baseline and 6-weeks after starting daily HCQ. Flow cytometry assays were designed to determine changes in T cell activation and T cell responses. Bead array multiplex were used to analyse antibodies and cytokine levels before and after HCQ intake.

### Results

As anticipated, HCQ treatment decreased *ex vivo* T cell activation. We observed a decrease in CD4$^+$CD161$^-$ expressing CCR5 (p = 0.015) and CD69 (p = 0.004) as well as in CD8$^+$CCR5$^+$ (p = 0.003), CD8$^+$CD161$^+$CCR5$^+$ (p = 0.002) and CD8$^+$CD161$^+$CD95$^+$ (p = 0.004). Additionally, HCQ decreased the proportion of Th17 expressing CD29 (p = 0.019), a subset associated with persistent inflammation. The proportion of T regulatory cells expressing the inhibitory molecule TIGIT was also reduced by HCQ (p = 0.003). As well, T cells from people on HCQ were less responsive to activation and cytokine production following stimulation with recall antigens and memory T cells were less likely to produce both IFNγ and TNFα following stimulation.

**Data Availability Statement:** The data used for this article can be found at https://doi.org/10.34990/FK2/ZZQ1QD.

**Funding:** KRF (OCH # 126275) from Canadian Institutes of Health Research (https://cihr-irsc.gc.ca/e/193.html). The funders had no role in study design, data collection and analysis, decision to publish, or preparation of the manuscript. JL (S5 386-01) from Grand Challenges Canada (https://www.grandchallenges.ca/). The funders had no role in study design, data collection and analysis, decision to publish, or preparation of the manuscript.

**Competing interests:** The authors have declared that no competing interests exist.

## Conclusion

This study shows HCQ is associated with lower T cell activation and decreased T cell cytokine production. While this study was not performed with the intent of looking at COVID vaccine response, it does provide important information about the changes in immune response that may occur in patient taking HCQ as a treatment for their autoimmune disease.

## Introduction

In March 2020, the World Health Organization (WHO) declared that the SARS-CoV-2 had become a pandemic. Currently, it has infected over 767 million people worldwide and killed more than 6 million people (WHO update). While COVID-19 infection can be asymptomatic or give mild symptoms, it can also lead to severe disease, hospitalization, and death (WHO). Despite being vaccinated against COVID-19, some individuals remain at higher risk of severe infections.

Pre-existing conditions such as HIV [1], chronic lung disease [2], heart conditions [3] or immunocompromised individuals [4] are at a higher risk of developing severe disease if they are infected by SARS-CoV-2. Studies have shown that people living with an immune-mediated inflammatory diseases (IMID), such as rheumatoid arthritis, systemic lupus erythematosus, or inflammatory bowel disease, can mount a good humoral response (according to the antibody level measured) after 2 doses of mRNA vaccine. However, a proportion of these individuals show a lower antibody response post vaccination compared to healthy adults [5] and a recent study showed that in people living with IMID, the waning of antibodies and T cell responses happen faster than in healthy controls [6]. Interestingly, in this study, the importance of the decline (antibody level and neutralization capacity) was associated with autoimmune disorder treatment [6]. Indeed, no SARS-CoV-2-specific immune response could be detected as early as 3 months post second dose in anti-tumor necrosis factor (TNF) treated patients, highlighting the importance of considering medication used in the policies to access extra vaccine doses [6].

Due to its effects in blocking inflammation, hydroxychloroquine (HCQ) is used to manage the symptoms of excessive immune activation observed in autoimmune diseases such as rheumatoid arthritis, Sjørgens syndrome, systemic lupus erythramatosus [7] and phospholipid antibody syndrome [8]. Its use has been largely discussed recently in the context of treatment [9–11] or pre-exposure prophylaxis against COVID-19 [12], where it was shown to be ineffective. However, little has been reported on how the use of HCQ as an immunomodulator in IMID patients could impact their broader immune response.

We conducted a HIV prevention proof-of-concept study to determine whether HCQ's anti-inflammatory activity would reduce levels of vaginal mucosa activated $CD4^+CCR5^+$ T lymphocytes, as these cells are the primary target cell for HIV infection. Detailed immune assessments of the female genital tract and blood were performed. While the impact of HCQ treatment on reducing mucosal $CD4^+CCR5^+$ lymphocyte levels was not as promising as the other drug investigated (acetylsalicylic acid-aspirin) [13], HCQ treatment was associated with several significant dampening effects in the systemic immune system that we are reported herein.

## Materials and methods

### Participants and study design

The Immune Quiescence (IIQ) study was a randomized, open-label pilot study conducted at the Pumwani Maternity and Baba Dogo community clinics in Nairobi, Kenya, in 2014 which

involved the recruitment of 104 HIV uninfected women. This study had two arms; 81 mg of ASA or 200 mg of HCQ taken daily for a six-week period. The main findings of the study have been published previously [13]. Data presented herein reflect the effect of HCQ treatment on the systemic immune system, which was not reported previously. Briefly, the inclusion criteria included: women aged between 18 and 55 years, presence of a uterus and cervix, willingness to take the study drug for 6-weeks, general good health, not taking any anti-inflammatory or immunosuppressive drugs, HIV seronegative, and having no history of cardiovascular diseases. Exclusion criteria included: having been pregnant in the last 12 months or breastfeeding, presence of a sexual transmissible disease (STI) at any time point during the study, menopause, taking medication that counteracts the study drugs, known allergy to the study drugs, history of heart burn, stomach pain, stomach ulcers, anemia, hemophilia, kidney or liver disease, psoriasis, glucose-6-phosphate-dehydrogenase deficiency, dermatitis, alcoholism, eye disease/ sight impairment, or being involved in another clinical trial. Ethics boards from the University of Nairobi/Kenyatta National Hospital and the University of Manitoba approved the study. Written informed consent was obtained from all participants.

## Enrolment and clinical procedures

Participants who met enrolment criteria and had normal liver and kidney function (creatinine, urea, sodium, potassium, aspartate amino transferase, alanine amino transferase, gamma-glutamyl transferase, alkaline phosphotransferase) were enrolled and randomized into one of the two study arms. Participant baselines were assessed at the first visit (BL), with samples collected prior to the initiation of the study drug, which they then took daily for six weeks. Participants returned to the clinic two weeks (Wk2) and six weeks (Wk6) later. At each study visit, blood, cervico-vaginal lavage (CVL), and endocervical cytobrush were taken and a questionnaire was filled out. Following the week 6 visit when the drug was stopped, a follow-up was scheduled 2 to 4 weeks later to reassess kidney and liver functions. In order to assess pill-taking adherence, participants were asked to return the pill container and a pill count was performed at each visit. Plasma and CVL were shipped to Winnipeg, Canada for drug-level measurement.

Clinical, demographic, and behavioural questionnaires were answered at every visit along with vaginal and blood samples. At each visit, a vulvovaginal swab was collected and used to assess the presence of Candida pseudohyphae and bacterial vaginosis by microscopy. *Trichomonas vaginalis* was diagnosed using the In-Pouch kit (Biomed Diagnostics, USA). Urine samples were collected for PCR detection of *Neisseria gonorrhea* and *Chlamydia trachomatis* (Roche Amplicor kits, USA). HIV serology using a rapid test (Determine, Inverness Medical, Japan) was performed at the first and last study visit. Women with a menstrual cycle were asked to come 5–8 days after bleeding stopped ensuring mucosal samples were taken during the follicular phase. Participant baseline (visit 1, BL) was used to assess systemic and mucosal immune activation prior to the initiation of drug therapy, making each participant their own control.

## Sample collection and processing

Blood was collected by venipuncture using heparin tubes. Peripheral blood mononuclear cells (PBMC) were isolated by Ficoll density gradient. T cell activation PBMCs was determined using flow cytometry in Nairobi, Kenya. Remaining PBMCs were frozen and shipped to Winnipeg, Canada, where stimulations and further characterization of the immune system took place.

## Drug level detection

HCQ levels in plasma were quantitated using reversed phase high-performance liquid chromatography (RP-HPLC) following a previously described method [14]. Drug contents in the samples were extracted and analyzed using the Symmetry® C18 column attached to a Waters® Alliance® HPLC system equipped with Waters® 2690 Separations module and Waters® 996 Photodiode Array detector. Chloroquine was used as an internal standard.

## IgG detection

Total IgG levels were determined using a Human IgG Subclass Profile kit (Invitrogen, Carlsbad, USA). Samples were run according to manufacturer's protocol at a dilution of 1:3000. Influenza specific IgG levels were assessed using an in-house bead array which has been previously described [15]. Briefly, antigens from 10 different influenza strains (H1N1 Soloman, H1N1 Puerto Rico, H3N2 Brisbane, H3N2 Hong Kong, H3N2 Kiev, H3N2 Wisconsin, H7N7 Netherlands, B Malaysia, B Quingdao, B Victoria) (Prospec, East Brunswick, New Jersey, USA) were selected. Antigens were bound to uniquely labelled fluorescent beads. Samples were run at a dilution of 1:100. Following incubation, beads were washed to remove unbound sample and exposed to biotinylated mouse anti-human IgG1. Influenza specific IgG was quantitated using streptavidin-PE and acquired on the Bioplex 200 suspension array system (BioRad, Mississauga, Ontario, Canada).

## Flow cytometry

Assays performed in Nairobi, Kenya were done on fresh cells. PBMC ($10^6$) and CMC (whole pellet) were washed with PBS containing 2% v/v FBS (FACS) and stained with the following antibodies: PE.Cy5-CD3, FITC-CD4, V500-CD8, PE-CD95, APC.H7-HLA-DR, APC-CD161, Alexa700-CD45RA, V450-CCR5, PE.Cy7-CD69, PE-CF594-CCR7 (BD Biosciences, USA), and Far Red-Live Dead discriminant (Invitrogen, USA). Data were acquired on an LSRII flow cytometer (BD System, USA) and analyzed using FlowJo v10.0.8r1 (BD Biosciences, San Jose, USA).

For frozen PBMCs, four separate flow cytometry assays were performed. The first panel was designed to assess T cell subsets and markers of cell trafficking. For this assay, cells were thawed and rested overnight, $10^6$ PBMCs were washed with 2% FBS-1x PBS and stained with APC-Fire-CD3, BV605-CD4, BV650-CD8, BV421-CD161, Alexa700-CD29, PE-Dazzle-CXCR3, PerCP-Cy5.5-Vα7.2 (BD Biosciences, USA), PE-CD103, PE-Cy7-CD25, BB515-CCR5, APC-CCR6 (Biolegend, San Diego, USA), and Aqua Vivid Dead discriminant (ThermoFisher, Waltham, USA). The second panel was designed to characterize T regulatory cells (Tregs). As before, cells were thawed and rested overnight, then $10^6$ PBMCs were washed with FACS and stained following a modified intracellular (nuclear antigen) protocol from ThermoFisher Scientific [16]. Antibodies used were as follows, PECy7-CD3, BV605-CD4, BV650-CD8, Alexa488-CD25, Alexa700-CD127, PE-Dazzle-TIGIT, APC-Fire-CD69, BV711-HLA-DR, PerCP-Cy5.5-CTLA-4 (Biolegend, San Diego, USA), PE-FoxP3, eFlour450--Helios (eBiosciences, Santa Clara, USA), and Aqua Vivid Dead discriminant (ThermoFisher, Waltham, USA).

Stimulations were performed on cryopreserved cells and two types of stimulations assays were performed, short term 12-hours and long-term 7 days. (1) CD3/CD28 beads (LifeTechnologies, USA) were used as a positive control (at a ratio of Beads:PBMCs of 1:1 for the 12-hour stimulation and 1:2 for the 7-day stimulation). (2) Two peptide pools were used for stimulation; Cytomegalovirus, Epstein Barr virus, Influenza—Flu (CEF) peptide pool (Cellular Technology Ltd, USA) at a concentration of 2μg/mL and a human papilloma virus (HPV)

peptide pool at a concentration of 8μg/mL that was designed based on a immune epitope mapping study using the E2 protein [17]. Ten immunodominant peptides were chosen and sent to ThermoFisher for custom peptide synthesis (ThermoFisher, USA), and mapped to the E2 protein of HPV16 as follows: E2-1 amino acids 30–44, E2-2 amino acids 94–108, E2-3 amino acids 99–114, E2-4 amino acids 136–150, E2-5 amino acids 143–157, E3-6 amino acids 150–164, E2-7 amino acids 286–301, E2-8 amino acids 301–315, E2-9 amino acids 309–323, E2-10 amino acids 335–350. PBMCs were thawed and rested overnight followed by stimulation for either 12 hours with Golgi Stop and Golgi Plug (BD Biosciences, USA) or pre-stained with CSFE (LifeTechnologies, USA) and stimulated for 7 days with feeding on day 5. All stimulations were conducted in RPMI+2% v/v inactivated human AB serum + 2% v/v Penicillin-Streptomycin (Sigma-Aldrich, USA). Following stimulation, cells were stained with BV711-CD3, BV605-CD4, BV650-CD8, APC-Fire-CD69, PECy7-CD95, PECy5-CD45RA, BV510-CCR7. Cells undergoing 12-hour stimulation were stained with PE-IL-2, BV421-IFNγ, Alexa488-TNFα, following intracellular permeabilization for cytokines according to manufacturer's protocol [16]. Cells undergoing 7-day stimulation were stained with APC-HLA-DR in addition to the phenotyping markers described above. For all four assays run in Winnipeg, Canada, data was acquired on an LSRII flow cytometer (BD System, USA) and analyzed using FlowJo v10.6.1 (BD Biosciences, USA).

## Statistical methods

Systemic T cell immune activation: The analyses performed for this pilot study included assessment of the difference in T cell immune activation between baseline and each visit separately. In the analysis, the $X^2$ test was used to assess the significance of the associations between categorical variables using Prism 6.0f (GraphPad Software, La Jolla, CA, USA). Gaussian distribution was tested by Shapiro-Wilk normality test and normality plot using SPSS (NY, USA). To compare baseline to the study visits, two-tailed paired T test or Wilcoxon matched-pair signed-rank test were performed. As this was a discovery study, p values were considered significant if ≤0.05. The study was monitored by a Data and Safety Monitoring Board derived from University of Manitoba. The study is registered on ClinicalTrials.gov (NCT02079077). Participants who were discontinued from the study were excluded from the analysis.

Cell trafficking, Treg and stimulation analyses: Data was cleaned using RStudio v3.5.0 (RStudio Inc, USA) and differences in the immune system before and after HCQ were compared using Wilcoxon match pair signed rank test in STATA v15.0 (StataCorp LLC, USA). Data was visualized using Prism v8.4.1(GraphPad Software, La Jolla, CA, USA). As this was a discovery study intention to treat analysis was performed, p values <0.05 were considered significant.

## Results

### Sociodemographics

Thirty-nine women were randomized into the HCQ arm. The average age was 30 years old. At baseline, 32 declared having a regular partner with a mean of sexual intercourse of 1.2 times in the last 7 days, all participants were enrolled in 2014. There was no difference between the number of regular partners, number of sex acts or condoms used between the study visits. The majority of women in this study were either on the injectable progestin-based contraception (depot medroxyprogesterone acetate—DMPA) (48%) or were not taking any hormonal contraception (41%). The type of contraception used was consistent during the course of the study. None of the participants on DMPA received their DMPA injection during the course of the study. Table 1 represents the data collected at baseline (BL).

**Table 1. Demographic factors reported at baseline and drug levels detected at baseline and week 6.**

|  | HCQ (n = 39) |
|---|---|
| Age, years (mean+/- SD) | 30+/-6 |
| Regular sex partner |  |
| Yes | 32 (82%) |
| no | 5 (12.8%) |
| not disclosed | 2 (5.1%) |
| Number of sexual intercourse with regular partner during the last 7 days (mean +/- SD) | 1.26 +/- 1.4 |
| Number of participants who used condom with regular partner during the last 7 days | 5 (12.8%) |
| Hormonal contraception used: (n (%)) |  |
| None | 16 (41.03%) |
| Injectable progesterone based (DMPA) | 19 (48.72%) |
| Oral pill | 3 (7.69%) |
| Other or not disclose | 1 (2.56%) |
| Plasma Drug level at baseline (mean +/- SD)[a] | Undetectable |
| Plasma drug level at week 6 (mean +/- SD)[a] | 37.1+/-18 |

n: number, SD: standard deviation

[a]limit of detection for HPLC based drug detection was 12ng/mL

## Antibody production

The capacity of the humoral response to produce antibodies is an important factor when generating a response to an infection or a vaccine. To determine the impact of HCQ on the systemic humoral immune response, total IgG and influenza-specific IgG levels were assessed over the 6-weeks of HCQ treatment. We observed that the level of plasma IgG significantly decreased over 6-weeks of HCQ treatment [median V1: 15136, median V3: 13378, (p = 0.006) (Fig 1A). However, there was no decrease in the level of influenza-specific IgG to seven influenza A and three influenza B isolates (Fig 1B).

## Ex vivo T cell phenotypes

T cells are an important cell of the adaptive immune system and their role is crucial for clearing viral infections [18]. It is therefore important to understand how HCQ impacts the T cell phenotype profile. We assessed the *ex vivo* profile of the T cell population in term of activation and trafficking markers to determine if HCQ treatment affects T cell activation and their capacity to migrate into the tissues. $CD4^+$ and $CD8^+$ T cell activation status was assessed on fresh cells, while Treg and expression of trafficking markers (CD25, CD29, and CCR6) on T cells were measured on cryopreserved cells.

Following 6-weeks on HCQ, the proportion of $CD4^+CD161^+CCR5^+$ (Th17-like cells) was lower compared to baseline [median BL: 26.05, median Wk 6: 13.2, p = 0.010]. The proportion of $CD4^+CD161^-$ expressing $CCR5^+$ [median BL: 17.85, median Wk 6: 8.82, p = 0.015] and $CD69^+$ [median BL: 7.44, median Wk 6: 4.28, p = 0.004] (Fig 2A) also decreased.

HCQ treatment also affected $CD8^+$ T cell activation. We observed that the percentage of $CD8^+ CCR5^+$ T cells decreased following HCQ treatment [median BL: 24.4, median Wk 6: 15.1, p = 0.003]. Furthermore, the proportion of $CD8^+CD161^+CCR5^+$ [median BL: 38.5, median Wk 6, 21.6, p = 0.002] and $CD8^+CD161^+CD95^+$ [median BL: 88.6, median Wk 6: 84, p = 0.004] significantly decreased compared to the percentage measured at baseline (Fig 2B).

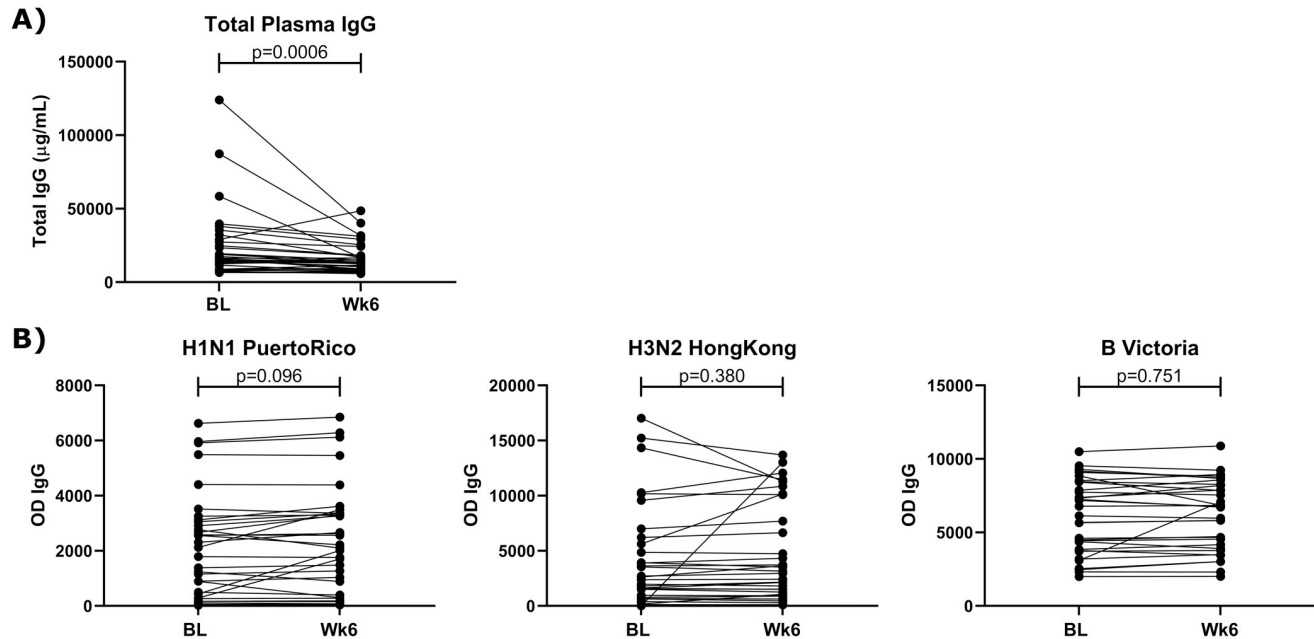

**Fig 1. A pair-wise comparison of the quantity of total IgG in the plasma at baseline (BL) and following for 6-weeks (Wk 6) of HCQ treatment.** A) Total IgG, B) select influenza specific isolates. Data was analyzed using Wilcoxon paired rank test and p values <0.05 were considered significant.

To further describe T cell populations, frozen PBMC were analyzed. An increase in the proportion of CD4$^+$ T cells [median V1: 37.8, median V3: 43.8, p = 0.047] as well as in the expression of CD4 on a per cell basis (Median Fluorescence Intensity—MFI) [median V1: 8090, median V3: 8516, p = 0.034] (Fig 2C) was observed. This is consistent with the data reported in Lajoie *et al.*, performed ex vivo on fresh PBMC [13].

In addition, following 6-weeks of HCQ treatment, Tregs (defined as CD4$^+$CD25$^+$FoxP3$^+$) expressing T cell Immunoglobulin and ITIM Domain (TIGIT) were lower compared to the proportion of these cells measured at baseline [median V1: 82.8, median V3: 76.6, p = 0.0025, p = 0.003]. The expression of TIGIT on a per cell basis was also lower after 6-weeks HCQ treatment [MFI: median V1: 3308.5, median B3: 2812.5, p = 0.022] (Fig 2C). HCQ treatment also decreased the percentage of the CD29$^+$ (integrin β1) Th17 cells (CD4$^+$CCR6$^+$CD161$^+$) [median V1: 91.7, median B3: 89.8, p = 0.019]. Finally, the expression of CCR6 on MAIT cells (CD4-CD161++Vα7.2+) [MFI: median V1: 1460, median V3: 1223, p = 0.011] and CD25 on Tc17 cells [median V1: 1213.5, median V3: 1190.5, p = 0.041] decreased following HCQ treatment compared to baseline (Fig 2C).

### T cell activation following in vitro antigen stimulation

In order to develop long-term immune responses, T cells must be able to generate immune memory and response to antigenic exposure [19–21]. Therefore, the functional capacities of memory T cells after 6-weeks of HCQ treatment were utilized a 12-hour or 7-day stimulation with CEF or HPV peptide pools (see data in Supporting Information). To assess the expression of three activation markers (acute activation marker CD69, the activation and pro-apoptotic marker CD95, and the chronic activation marker HLA-DR). This analysis was carried out on bulk CD4$^+$ and CD8$^+$ T cells as well as naïve (CD45RA$^+$CCR7$^+$), central memory

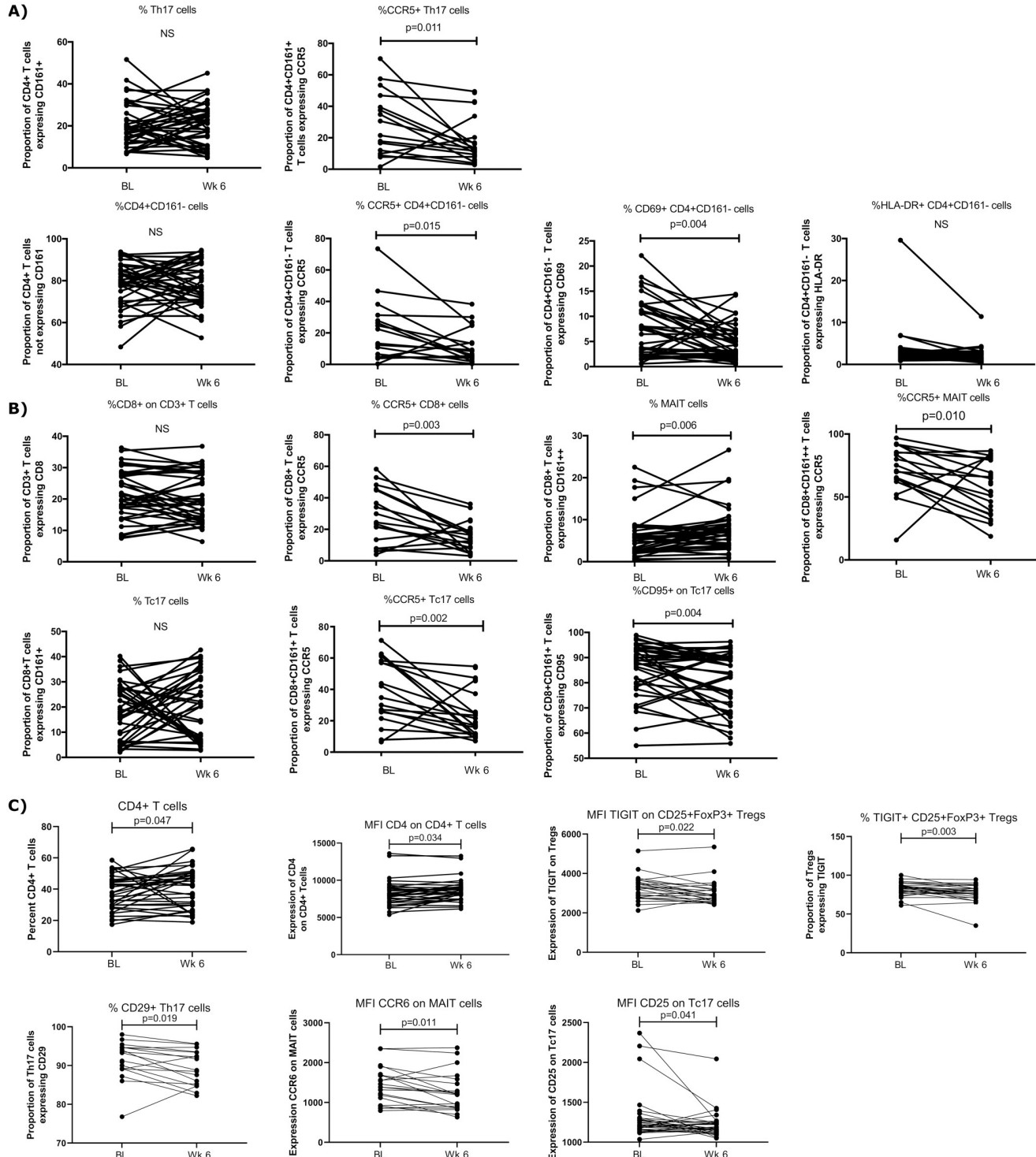

**Fig 2. Effect of HCQ on T cell subsets from *ex vivo* PBMCs.** Data were analyzed using Wilcoxon paired rank test and p values <0.05 were considered significant. A) Changes in the proportion of CD4+ T cells in the fresh blood of participants treated with HCQ for 6-weeks. B) Changes in the proportion of CD8+ T cells in the fresh blood of participants treated with HCQ for 6-weeks. C) Changes in the proportion of T cell markers in the cryopreserved blood of participants treated with HCQ for 6-weeks. T regulatory cells (Tregs), Mucosal associated invariant T cells (MAIT). Data was analyzed using Wilcoxon paired rank test and p values <0.05 were considered significant.

(CD45RA⁻CCR7⁺), and effector memory (CD45RA⁻CCR7⁻) T cells. Only data from responder to the peptide pools are presented in the following sections.

**Twelve-hour stimulation.** In the CD4⁺ T cell compartment, following 12-hour stimulation with the CEF peptide pool, there was a decrease in the overall proportion [median V1: 53.8, median V3: 47.5, p = 0.035] and expression of CD95 on CD4⁺ T cells [median V1: 1439.5, median V3: 1381.5, p = 0.001] following HCQ treatment. In the memory cell subsets, we observed a decreased in the expression of CD95 on central memory CD4⁺ T cells [MFI median V1: 1568.5, MFI median V3: 1546.5, p = 0.015], effector memory CD4⁺ T cells [MFI median V1: 1457.5, MFI median V3: 1387.5, p = 0.001], and naïve CD4 T cells [MFI median V1: 1434.5, MFI median V3: 1336, p = 0.033]. Furthermore, the proportion of CD95+ on naïve CD4⁺ T cells decreased post HCQ [median V1: 13.8, median V3: 10.45, p = 0.003] (Fig 3A).

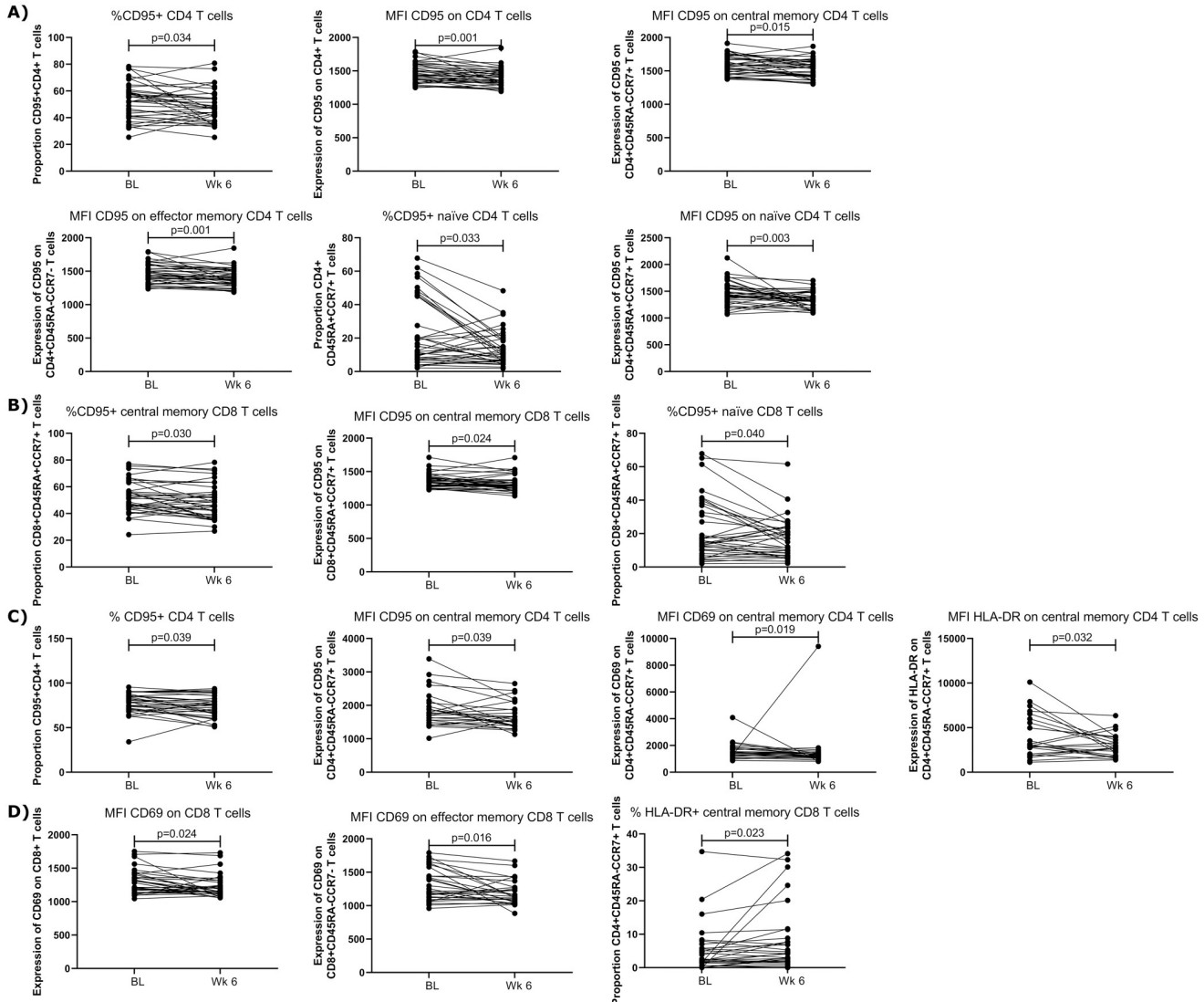

**Fig 3. Effect of HCQ on T cell activation markers post stimulation with 2μg/mL CEF (cytomegalovirus, Epstein Barr virus, and influenza virus) peptide pool.** A) Changes in CD4+ T cell markers following 12-hour stimulation. B) Changes in CD8+ T cell markers following 12-hour stimulation. C) Changes in CD4+ T cell markers following 7-day stimulation. D) Changes in CD8+ T cell markers following 7-day stimulation with CEF. Data was analyzed using Wilcoxon paired rank test and p values <0.05 were considered significant.

Similar decreases in the expression or proportion of CD95 on CD4+ T cells were seen following stimulation with the HPV peptide pool (S1A Fig).

In the CD8[+] T cell compartment, following 12-hour CEF peptide pool stimulation, the proportion of central memory CD8[+]CD95[+] T cells [median V1: 51.45, median V3: 47.45, p = 0.030] as well as its expression per cell [median V1: 1342, median V3: 1297.5, p = 0.024] decreased with HCQ treatment. We also observed an increase in the proportion of naïve CD8[+]CD95[+] T cells [median V1: 15.1, median V3: 16.05, p = 0.040] post HCQ (Fig 3B).

**Seven-day stimulation.** In line with the 12-hour assay, following CEF peptide pool stimulation the proportion of CD95 on bulk CD4[+] T cells [median V1: 78.3, median V3: 76, p = 0.039] and the expression of CD95 on central memory CD4[+] T cells [median V1: 1786, median V3: 1661, p = 0.039] decreased post HCQ. We also observed a decrease in the expression of CD69 [median V1: 1432, median V3: 1299, p = 0.019] and HLA-DR [median V1: 3109.5, median V3: 2707.5, p = 0.032] on central memory CD4[+] T cells after HCQ treatment (Fig 3C). Similar decreases in HLA-DR were seen in effector memory CD4 T cells following HPV peptide pool stimulation (S1B Fig).

When looking at the impact of CEF stimulation on the CD8[+] T cell compartment, we observed a decrease in the expression of CD69 on bulk CD8[+] T cells [median V1: 1284, median V3: 1188, p = 0.024] and on effector memory CD8[+] T cells [median V1: 1213, median V3: 1165, p = 0.016] post-HCQ. CEF stimulation also resulted in an increase proportion of HLA-DR on central memory CD8[+] T cells [median V1: 2.29, median V3: 4.195, p = 0.023] (Fig 3D). Similar decreases in CD69 were seen in central memory and naive CD8 T cells following HPV peptide pool stimulation (S1C Fig).

## Impact of HCQ on T cell capacity to produce cytokine after stimulation

Cytokines are the chemical messengers by which immune cells affect their function. Therefore, it is important to assess whether the immune cells capacity to produce cytokines following stimulation is altered by HCQ treatment. The effect of HCQ treatment on cytokine production (IFNγ, TNFα, IL-2) was assessed in three ways: the total amount of cells responding to stimulation, the amount of cytokine produced on a per-cell basis, and the ability of cells to co-express the three cytokines.

When determining overall concentration of cytokines produced, we used an adapted version of the background corrected cytokines [19]. Herein, cytokines detected in the unstimulated conditions were subtracted from stimulation conditions (CEF or HPV peptide pools) to allow assessment of the effect of HCQ on cytokine production in responding individuals. Overall, HCQ treatment decreased the amount of cytokine produced. As seen in Fig 4A, following stimulation with the CEF peptide pool, the percent of TNFα decreased in both CD4[+] T cells [median V1: 0.5, median V3: 0.25, p = 0.003] and effector memory CD4[+] T cells [median V1: 0.39, median V3: 0.16, p = 0.030]. There is also a decrease in the percent of IL-2 produced by naïve CD8[+] T cells following stimulation with the CEF peptide pool [median V1:0.104, median V3:0.044, p = 0.031] which was also seen with the HPV peptide pool (S2A Fig).

As stated above, the amount of cytokine produced on a per cell basis was also assessed by determining the median fluorescence intensity (MFI) of each cytokine after stimulation. In general, the amount of cytokine produced on a per cell basis was decreased following HCQ treatment. Background IFNγ production (unstimulated condition) decreased in bulk CD4[+] T cells [median V1: 2064, median V3: 1886, p = 0.0012], central memory CD4[+] T cells [median V1: 2091, median V3: 1879, p = 0.0009], naïve CD4[+] T cells [median V1: 2091, median V3: 1989, p = 0.0153], and central memory CD8[+] T cells [median V1: 2083.5, median V3: 1953, p = 0.0040] (Fig 4B). Stimulation with CEF peptide pool resulted in decreased expression of

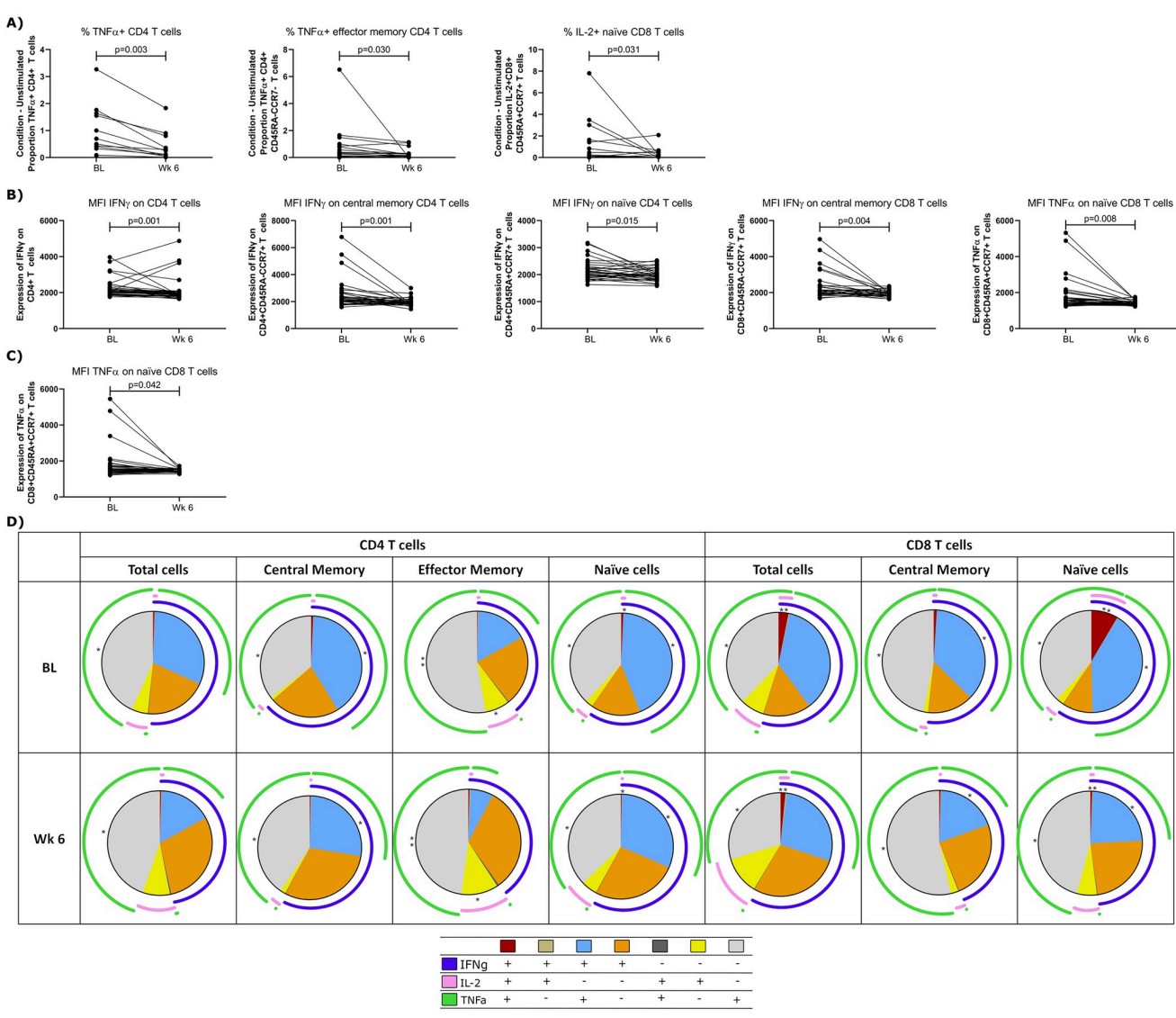

**Fig 4. Effect of HCQ on cytokines after a 12-hour stimulation with 2µg/mL CEF (cytomegalovirus, Epstein Barr virus, and influenza virus) peptide pool.** A) Cytokine production determined by subtracting the amount of cytokine detected in the unstimulated condition from each of the stimulated conditions used in this study, B) Cytokine expression in unstimulated cells, C) cytokine expression following stimulation with the CEF peptide pool. D) Co-expression of multiple cytokines, p<0.05 (*), p<0.01 (**), pie colours represent different co-expression options, arc legend colours indicate the three cytokines assessed. Data was analyzed using Wilcoxon paired rank test and p values <0.05 were considered significant.

TNFα on naïve CD8+ T cells [median V1: 1506, median V3: 1453, p = 0.042] (Fig 4C) though it should be noted there was also decrease in TNFα expression in naïve CD8+ T cells in the absence of stimulation (unstimulated cells) [median V1: 1487, median V3: 1425, p = 0.0084] (Fig 4B). Stimulation with the HPV peptide pool resulted in decreases in the expression of cytokines (particularly IFNγ on CD4+ T cells and TNFα on CD8+ T cells) following HCQ treatment (S2B Fig).

Finally, the effect of HCQ on the co-expression of IFNγ, TNFα, and IL-2 on T cell subsets was assessed. The Boolean gating option in FlowJo, which allows observation of all combinations of co-expression for chosen markers, was used. Pie charts of median numbers of cells

expressing at least one cytokine to all three cytokines in response to 12-hour stimulation with the CEF pool can be observed in Fig 4D. Following CEF peptide pool stimulation in all T cell subsets, except effector memory CD8+ T cells, decreases in T cells co-expressing IFNγ and TNFα were found with corresponding increases in the number of cells expressing only one of the cytokines. In addition, for total CD8+ T cells and naïve CD8+ T cells, there was a significant decrease in the number of cells co-expressing all three cytokines (Fig 4D). Following HPV peptide pool stimulation decreased TNFα and IL-2 co-expression was observed (S2C Fig).

While cytokine responses provides a measure of immediate functional ability in response to an infectious agent, the ability of immune cells to expand and proliferate in response to stimulation is crucial to combating the infection [22]. Therefore, during our 7-day stimulation assay, we assessed the effect of HCQ on the T cell capacity to proliferate. While cellular division occurred, HCQ had little to no effect on the percent of cells able to divide when stimulated with 3 different conditions (CEF, HPV, CD3/CD28 beads) (S3 Fig).

## Discussion

COVID-19 has brought forward many challenges and highlighted that there is still a lack of knowledge regarding how the human immune system responds to viral infection and vaccines. One of these gaps in knowledge is how certain medications may impact the capacity of the immune system to respond to viral infection or vaccines. People suffering from autoimmune diseases like rheumatoid arthritis or systemic lupus erythematosus are often being prescribed HCQ to manage their conditions [7], but little is known about how this impacts their capacity of establishing an immune response to viral infection or vaccines. Based on pilot study results from 2014, in the context of preventing HIV infection in healthy participants, we are reporting herein some of our findings on how HCQ modifies the systemic immune response toward a less activated and less responsive phenotype. This is important as it could impact how people treated with HCQ respond to immune challenges.

With the COVID vaccine roll-out in 2021, we saw an important reduction in SARS-CoV2 infections and infections related-morbidity and mortality [23]. However, with the emergence of new variants, we also observed a decrease in the neutralization capacity of the antibodies produce by the vaccines [24, 25]. Since the start of the pandemic, most of the focus (both for vaccine development and immunity surveillance) has been put toward the capacity of individuals to produce antibodies [26–31]. The decline in antibodies produced in response to infection or vaccine over time was observed at a faster rate in some individuals with immune disorders and could be associated with treatments they were taking [6].

Understanding if a potential treatment can impact the humoral response against the virus is therefore highly relevant. HCQ has been shown to reduce the level of antiphospholipid antibodies [32]. In Sjögren syndrome patients, the use of HCQ decreases the level of IgG and IgM [33]. In our study, we observed that 6-week HCQ treatment significantly decreases the level of total IgG in healthy individuals. This was previously observed in HIV infected individuals who took HCQ [34]. Though our bulk immunoglobulin assay could not indicate which type of IgG was decreased, in a second more specific assay we observed that the level of influenza-specific IgG was not changed by the use of HCQ. This is important because despite an overall decrease in the blood IgG titer, the level of specific viral IgG was not affected. It is encouraging to note that the influenza-specific antibodies did not wane and may suggest that pre-existing antibodies formed prior to HCQ may not be affected. However, the decrease in total IgG levels raises concerns if, for those taking HCQ, their *de novo* responses to newly delivered vaccines would be as effective.

While the presence of antibodies is an important measure of responsiveness to vaccines or infection, such as SARS-CoV-2, there is mounting evidence showing that T cell responses are required for broad and durable protection [19–21]. A study from Grifoni et al. showed that circulating SARS-CoV-2 specific CD8[+] and CD4[+] T cells could be observed in 70% and 100%, respectively, of COVID-19 convalescent patients [20]. Furthermore, it was shown that memory T cells against SARS-CoV-1 could be detected 17 years after exposure, which highlights the importance that T cells could have on the duration of protection [35].

T cells play an important role in rheumatoid arthritis pathology. The presence of memory CD4[+] T cells in joints, the frequencies of circulating Th17 and serum levels of IL-17 are associated with disease activity [36–39]. HCQ has been shown to interfere with the intracellular calcium stores required for T cell receptor stimulation resulting in decreased expression of the early activation marker CD69 [40]. It has also been observed that it inhibits CD4[+] T cell activation by AP-1 signaling modulation [41]. Herein, we observed an increase in bulk circulating CD4[+] T cells, as well as, the expression of CD4 receptor per cell after 6-weeks on HCQ treatment compared to baseline. Despite this increase, HCQ treatment led to a decrease in the proportion of CD4[+]CD161[-] expressing CCR5 or CD69, as well as CD8[+]CCR5[+] and CD8[+]C161[+] expressing CCR5 or CD95. Our results are supported by a study from Goldman et al., showing that HCQ inhibited anti-TCR-induced-up-regulation of CD69 expression of CD4[+] T cells [40] T cells expressing CCR5 have been suggested to play an important role in the rheumatoid arthritis pathology due to their capacity to migrate to inflamed synovium [42]. The capacity of HCQ treatment to decrease the expression of CCR5 was also observed in a study from Hachim et al [43]. Interestingly, CD4[+]CD29[+] T cells have been associated with the persistent inflammation observed in people with ulcerative colitis [44]. In this study, we showed that the proportion of Th17 expressing CD29 was decreased after 6-weeks on HCQ. Th17 are a CD4[+] T cell subset that produce a high quantity of IL-17 [45] and have an important role in inflammatory responses and autoimmune diseases [46]. Together, our data shows that HCQ treatment decreased ex vivo T cell activation in our population of healthy donors. While this decrease of T cell activation is beneficial for people suffering from rheumatoid arthritis or systemic lupus erythematosus, it could impact their ability to mount a proper immune response to viral infections or vaccines.

To determine if HCQ treatment affects T cell capacity to respond to recall antigen or to produce cytokines, we stimulated the PBMC of our participants obtained at baseline and after 6-weeks on HCQ with peptide pools to produce a recall antigen response. The results showed that after stimulation (short-term 12-hour or long-term 7-day assays), the expression of CD95 on memory T cells was decreased. After the longer stimulation there was also a decrease in the expression of the early activation marker CD69 on central memory CD4[+] T cells and effector memory CD8[+] T cells. Furthermore, the expression of the chronic activation marker HLA-DR was also affected on central memory (both CD4[+] and CD8[+]) and on effector CD4+ T cell memory. When looking at T cell capacity to produce cytokines after stimulation of PBMC collected before and after HCQ uptake, we observed that the expression of TNFα from bulk CD4[+] T cells and effector CD4[+] T cells was affected. Indeed, those cell subsets produced less TFNα after 6-weeks on HCQ. The production of TFNα was also affected for CD8[+] naïve T cells. By looking at cytokine expression with Boolean gating, we also observed that the production of cytokines after treatment on HCQ was less diverse. In all the T cell populations studied, the population co-expressing both TFNα and IFNγ decreased. These data show that when exposed to previously exposed antigens, the capacity of the T cell memory compartment to get activated to mount an immune response is affected by HCQ treatment.

Our study has some limitations that must be mentioned. The duration of time on HCQ was only 6-weeks which is very short compared to people taking HCQ for treating autoimmune disease. This study was not designed to look at how HCQ impacts the immune response to

SARS-CoV-2 infection or the response to vaccines against COVID-19, as it was conducted in 2014. However, it does offer some insight about how HCQ treatment may result in an altered immune response which may impact responses to respiratory and mucosal viruses. Those changes, which include reductions in T cell memory immune response and a less diverse cytokine production in response to stimulation with respiratory and mucosal recall antigens, may be important factors for *de novo* responses to other infections or vaccines.

## Conclusions

Our findings provide information about the importance of understanding how HCQ treatment affects the immune system. Herein 6-week 200mg/day reduced T cell activation and cytokine production to peptide pools indicating a dampened immune response. This could have implications for people suffering from immune disorders who use HCQ treatment.

## Supporting information

**S1 Fig. Effect of HCQ on T cell activation markers post stimulation with 8μg/mL HPV (human papilloma virus) peptide pool.** A) Changes in CD4+ T cell markers following 12-hour stimulation. B) Changes in CD4+ T cell markers following 7-day stimulation. C) Changes in CD8+ T cell markers following 7-day stimulation. Data was analyzed using Wilcoxon paired rank test and p values <0.05 were considered significant.
(TIF)

**S2 Fig. Effect of HCQ on cytokines after a 12-hour stimulation with 8μg/mL HPV (human papilloma virus) peptide pool.** A) Cytokine production determined by subtracting the amount of cytokine detected in the unstimulated condition from each of the stimulated conditions used in this study, B) cytokine expression. C) Co-expression of multiple cytokines, p<0.05 (*), p<0.01 (**), pie colours represent different co-expression options, arc legend colours indicate the three cytokines assessed. Data was analyzed using Wilcoxon paired rank test and p values <0.05 were considered significant.
(TIF)

**S3 Fig. Effect of HCQ on proliferative ability after a 7-day stimulation.** A) Proliferation following stimulation with 2μg/mL CEF (cytomegalovirus, Epstein Barr virus, and influenza virus) peptide pool. B) Proliferation following stimulation with 8μg/mL HPV (human papilloma virus) peptide pool cytokine expression. Data was analyzed using Wilcoxon paired rank test and p values <0.05 were considered significant.
(TIF)

## Acknowledgments

The authors thank all the participants of the IIQ study as well as the female sex workers from the Pumwani and Baba Dogo cohort for their support.

## Author Contributions

**Conceptualization:** Julie Lajoie, Keith R. Fowke.

**Formal analysis:** Monika M. Kowatsch, Julie Lajoie.

**Funding acquisition:** Julie Lajoie, Keith R. Fowke.

**Investigation:** Monika M. Kowatsch, Julie Lajoie.

**Methodology:** Monika M. Kowatsch, Julie Lajoie, Lucy Mwangi, Kenneth Omollo, Natasha Hollett.

**Project administration:** Julie Lajoie, Julius Oyugi, Joshua Kimani.

**Visualization:** Monika M. Kowatsch, Julie Lajoie.

**Writing – original draft:** Monika M. Kowatsch, Julie Lajoie.

**Writing – review & editing:** Monika M. Kowatsch, Julie Lajoie, Lucy Mwangi, Kenneth Omollo, Julius Oyugi, Natasha Hollett, Joshua Kimani, Keith R. Fowke.

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
