## [Decision Letter · Decision Letter 0]

14 Feb 2023

PONE-D-22-33598

Hydroxychloroquine reduces T cells activation recall antigen responses: Implications for COVID-19 prevention

PLOS ONE

Dear Dr. Fowke,

Thank you for submitting your manuscript to PLOS ONE. After careful consideration, we feel that it has merit but does not fully meet PLOS ONE’s publication criteria as it currently stands. Therefore, we invite you to submit a revised version of the manuscript that addresses the points raised during the review process.

ACADEMIC EDITOR: 

I agree with comments of both reviewers, however, please ignore the comment #2 of the reviewer #2.

Additionally, please separate the figure legend from the main text. It would be more appreciated if the figure legends are stated in a separate page.

We look forward to receiving your revised manuscript.

Kind regards,

Janin Nouhin, Ph.D.

Academic Editor

PLOS ONE

Journal Requirements:

"Funding support was provided by the Canadian Institutes of Health Research (CIHR) (OCH # 126275) and S5 386-01 from Grand Challenge Canada (JL). "

"KRF (OCH # 126275) from Canadian Institutes of Health Research (https://cihr-irsc.gc.ca/e/193.html). The funders had no role in study design, data collection and analysis, decision to publish, or preparation of the manuscript.

JL (S5 386-01) from Grand Challenges Canada (https://www.grandchallenges.ca/). The funders had no role in study design, data collection and analysis, decision to publish, or preparation of the manuscript."

Reviewers' comments:

Reviewer's Responses to Questions

**Comments to the Author**

1. Is the manuscript technically sound, and do the data support the conclusions?

Reviewer #1: Yes

Reviewer #2: Yes

2. Has the statistical analysis been performed appropriately and rigorously? 

Reviewer #1: Yes

Reviewer #2: Yes

3. Have the authors made all data underlying the findings in their manuscript fully available?

Reviewer #1: Yes

Reviewer #2: Yes

4. Is the manuscript presented in an intelligible fashion and written in standard English?

Reviewer #1: Yes

Reviewer #2: Yes

5. Review Comments to the Author

Reviewer #1: Summary :

This study characterizes T cell responses in healthy individuals administered with HCQ over a period of six weeks. The authors demonstrate decreased T cell activation in samples taken at treatment endpoint compared to baseline. In addition, T cells obtained from these individuals after treatment with HCQ seem to be refractory to short or long stimulation with antigen peptides, as evidenced by their impaired cytokine production. All of these observations are in line with previous literature on the impact of HCQ on T cell responses. Altogether, the authors use the results of this study to advocate for better monitoring of SARS-CoV-2 vaccine responses in individuals on long-term HCQ treatment regimens.

General comments :

- There is some literature to suggest that HCQ induces apoptosis of T cells. Therefore, did the authors observe any differences in absolute counts of total T cells, CD4+ or CD8+ T cells or percentages of live/dead cells within the populations gated for T cells, especially at 6 wks post treatment.

- Since CD69 is an early activation marker, did the authors look at its expression on T cells in the 12-hour stimulation conditions? These findings could be included in the manuscipt.

- In the methodology section, the authors describe an HPLC based method to determine concentrations of HCQ in patient samples. The HCQ concentrations at baseline and Wk. 6 could be included in Table 1. Furthermore, did the concentrations of HCQ correlate with CD4 and CD8 T cells at baseline and Wk. 6?

Minor comments :

- Materials and methods could be modified to only include methodologies described in the results, e.g. fresh

cervical mononuclear cells were not described in the analyses.

- Line 250 : Should read “Figure 2C” instead of “Figure 3C”

- Resolution of the figure images should be improved.

Reviewer #2: In this study, the author highlights that hydroxychloroquine (HCQ) treatment used for patients suffering with autoimmune disorders may affect their immune response by reducing the T-cell activation and cytokine production.

Major issues:

1. This study elegantly establishes the relationship between HCQ and immune response, although, no evidence has been provided on HCQ affecting the efficacy of COVID vaccine. Hence, all the text in the manuscript must be corrected to make this point clear.

2. Additional validation studies maybe performed to study the levels of T-cell reduction over time.

3. Result section needs to modified extensively. The author needs to explain the rationale and the interpretation of their observed results.

4. In the discussion section, the author mentions that their findings help understand “how long-term treatment used by people suffering from immune disorders may impact their immune response”. Although, this study has been performed for 6 weeks, so it cannot be called as “long-term” in this manuscript.

Minor issues:

1. Table. 1 needs to be well structured and formatted.

2. Title of the manuscript needs to be reframed appropriately.

Overall, a good attempt at understanding the effect of HCQ treatment on immune response.

6. PLOS authors have the option to publish the peer review history of their article (what does this mean?). If published, this will include your full peer review and any attached files.

Reviewer #1: No

Reviewer #2: No

---

## [Author Response · Author response to Decision Letter 0]

13 Mar 2023

Thank you very much for your email of Feb 14, 2023 inviting us to resubmit a revised version of our manuscript: Hydroxychloroquine reduces T cells activation recall antigen responses: Implications for COVID-19 prevention. The suggestions of the reviewers and editorial team have improved the manuscript and we thank them for their comments. We can find below our detailed response to the comments. A revised manuscript with changes tracked has been provided as has an unmarked version. We hope that we have sufficiently addressed the items noted and that the manuscript will be acceptable to the reviewers and editors. We look forward to our manuscript being published in PLOS One.

Figure resolution has been checked using the journal’s PACE system, please click on the link in the top right of the figures to download a high resolution version of each figure. 

All information below is also available in the attached "response to reviewer document". AU denotes author responses to reviewer comments.

In response to the comments from the academic editor and journal requirements

We appreciate that the editor indicated that additional experiments are not required.

Additionally, please separate the figure legend from the main text. It would be more appreciated if the figure legends are stated in a separate page.

AU: All figure legends have been moved to 1 page following the reference section.

2. Please ensure that your manuscript meets PLOS ONE’s style requirements, including those for file naming. The PLOS ONE style templates can be found at

AU: The submitted file has been updated to match the above style guides. 

“Funding support was provided by the Canadian Institutes of Health Research (CIHR) (OCH # 126275) and S5 386-01 from Grand Challenge Canada (JL). “

“KRF (OCH # 126275) from Canadian Institutes of Health Research ( https://cihr-irsc.gc.ca/e/193.html). The funders had no role in study design, data collection and analysis, decision to publish, or preparation of the manuscript.

JL (S5 386-01) from Grand Challenges Canada ( https://www.grandchallenges.ca/). The funders had no role in study design, data collection and analysis, decision to publish, or preparation of the manuscript.”

AU: Funding information has been removed from the acknowledgements section. The current funding statement (as stated above) is correct.

a) If there are ethical or legal restrictions on sharing a de-identified data set, please explain to them in detail (e.g., data contain potentially sensitive information, data are owned by a third-party organization, etc.) and who has imposed them (e.g., an ethics committee). Please also provide contact information for a data access committee, ethics committee, or other institutional bodies to which data requests may be sent.

AU: We believe in the importance of data sharing. We also recognize the importance of respectful engagement with our community partners. Through the University of Manitoba we will generate a de-identified data set which will be deposited on a publicly accessible data repository platform. This will be communicated with our community partner.

In response to the comments from reviewer 1

This study characterizes T cell responses in healthy individuals administered with HCQ over a period of six weeks. The authors demonstrate decreased T cell activation in samples taken at treatment endpoint compared to baseline. In addition, T cells obtained from these individuals after treatment with HCQ seem to be refractory to short or long stimulation with antigen peptides, as evidenced by their impaired cytokine production. All of these observations are in line with previous literature on the impact of HCQ on T cell responses. Altogether, the authors use the results of this study to advocate for better monitoring of SARS-CoV-2 vaccine responses in individuals on long-term HCQ treatment regimens.

General comments :

There is some literature to suggest that HCQ induces apoptosis of T cells. Therefore, did the authors observe any differences in absolute counts of total T cells, CD4+ or CD8+ T cells or percentages of live/dead cells within the populations gated for T cells, especially at 6 wks post treatment.

AU: Between baseline and Wk6 we did not see differences in the proportion of T cells. As we did not use counting beads during flow cytometry, getting a true absolute count is not possible. However, we did count the cells using trypan blue vitality stain and there was no difference in cell viability between baseline and week 6.

Since CD69 is an early activation marker, did the authors look at its expression on T cells in the 12-hour stimulation conditions? These findings could be included in the manuscript.

AU: CD69 was assessed at the 12-hour stimulation condition however no difference was found between baseline and week6.

In the methodology section, the authors describe an HPLC based method to determine concentrations of HCQ in patient samples. The HCQ concentrations at baseline and Wk. 6 could be included in Table 1. Furthermore, did the concentrations of HCQ correlate with CD4 and CD8 T cells at baseline and Wk. 6?

Plasma drug levels have now been included in Table 1 for plasma. No drug was detected at baseline for any participant therefore no correlations were performed for this time point.

AU: Thank you for the suggestion to perform correlation analyses. We did so on week 6 data, however, none of the major T cell markers showed strong correlations and we felt it did not add much to the paper so did not include that data. 

Minor comments :

Materials and methods could be modified to only include methodologies described in the results, e.g. fresh cervical mononuclear cells were not described in the analyses.

AU: The mention of cervical mononuclear cell sample collection and processing has been removed.

Line 250 : Should read “Figure 2C” instead of “Figure 3C”

AU: Thank you for the comment, the figure number has been corrected.

Resolution of the figure images should be improved. 

AU: Thank you for the comment, the resolution has been maximized.

In response to the comments from reviewer 2

In this study, the author highlights that hydroxychloroquine (HCQ) treatment used for patients suffering with autoimmune disorders may affect their immune response by reducing the T-cell activation and cytokine production.

Major issues:

1. This study elegantly establishes the relationship between HCQ and immune response, although, no evidence has been provided on HCQ affecting the efficacy of COVID vaccine. Hence, all the text in the manuscript must be corrected to make this point clear.

AU: Thank you for this comment, we agree that we must be clearer in our messaging. To reflect this comment, we have modified the title to remove any reference to COVID-19. We have also made modifications throughout the paper to clarify this issue.

2. Additional validation studies maybe performed to study the levels of T-cell reduction over time

AU: Thank you for this comment, we will be following the editor’s instruction to not perform additional studies.

3. Result section needs to modified extensively. The author needs to explain the rationale and the interpretation of their observed results.

AU: In the result section we have added additional information to ensure that there is sufficient rationale described for each section. We have utilized the discussion section for focusing on the interpretation of the results. 

4. In the discussion section, the author mentions that their findings help understand “how long-term treatment used by people suffering from immune disorders may impact their immune response”. Although, this study has been performed for 6 weeks, so it cannot be called as “long-term” in this manuscript.

AU: We agree with the reviewer’s statement and the conclusion has been modified accordingly.

Minor issues:

1. Table. 1 needs to be well structured and formatted.

AU: Reformatted for single space with no blank cells, table legend and title formatted according to the style guide.

2. Title of the manuscript needs to be reframed appropriately.

AU: We agree, the title has been changed. Our new title is “Hydroxychloroquine reduces T cells activation recall antigen responses”

---

## [Decision Letter · Decision Letter 1]

13 Jun 2023

Hydroxychloroquine reduces T cells activation recall antigen responses

PONE-D-22-33598R1

Dear Dr. Fowke,

We’re pleased to inform you that your manuscript has been judged scientifically suitable for publication and will be formally accepted for publication once it meets all outstanding technical requirements.

Kind regards,

Janin Nouhin, Ph.D.

Academic Editor

PLOS ONE

Additional Editor Comments (optional):

Please address the minor comment of reviewer #2:

Minor comments:

1. If there is any new information added from previous studies in the revised manuscript, the references for that must be added.

Please address the minor comment of reviewer #3:

Minor suggestions:

1. In Sociodemographics section, I suggest adding a sentence when recruitment was conducted. This will help put the study in context with regard to SARS-CoV-2 pandemic.

2. Line 363. I would use a different phrase or word for "uptake."

3. Line 402. I would use a different phrase or word for "in take."

4. Line 429-432. A sentence that is specific would be helpfull in the Conclusions section.

Reviewers' comments:

Reviewer's Responses to Questions

**Comments to the Author**

1. If the authors have adequately addressed your comments raised in a previous round of review and you feel that this manuscript is now acceptable for publication, you may indicate that here to bypass the “Comments to the Author” section, enter your conflict of interest statement in the “Confidential to Editor” section, and submit your "Accept" recommendation.

Reviewer #1: All comments have been addressed

Reviewer #2: All comments have been addressed

Reviewer #3: (No Response)

2. Is the manuscript technically sound, and do the data support the conclusions?

Reviewer #1: Yes

Reviewer #2: Yes

Reviewer #3: Yes

3. Has the statistical analysis been performed appropriately and rigorously? 

Reviewer #1: Yes

Reviewer #2: Yes

Reviewer #3: Yes

4. Have the authors made all data underlying the findings in their manuscript fully available?

Reviewer #1: Yes

Reviewer #2: Yes

Reviewer #3: Yes

5. Is the manuscript presented in an intelligible fashion and written in standard English?

Reviewer #1: Yes

Reviewer #2: Yes

Reviewer #3: Yes

6. Review Comments to the Author

Reviewer #1: (No Response)

Reviewer #2: Major comments:

1. The title of the manuscript has been appropriately modified as per suggestion.

2. All the comments for conclusion and result sections have been appropriately addressed.

3. Table. 1 has been accurately revised.

Minor comments:

1. If there is any new information added from previous studies in the revised manuscript, the references for that must be added.

The author has made all the necessary changes based on suggestions from all the reviewers.

Reviewer #3: Kowatsche et al. have presented findings that HCQ affects immune recall to specific antigens in a study conducted over a period of 6 weeks. The resuls are convincing and the summary and conclusions sound. The authots have responded adequatly to previous reviews. I have only minor suggestions.

Minor suggestions:

1. In Sociodemographics section, I suggest adding a sentence when recruitment was conducted. This will help put the study in context with regard to SARS-CoV-2 pandemic.

2. Line 363. I would use a different phrase or word for "uptake."

3. Line 402. I would use a different phrase or word for "in take."

4. Line 429-432. A sentence that is specific would be helpfull in the Conclusions section.

7. PLOS authors have the option to publish the peer review history of their article (what does this mean?). If published, this will include your full peer review and any attached files.

Reviewer #1: No

Reviewer #2: No

Reviewer #3: **Yes: **David Joseph Kelvin

---

## [Editor Report · Acceptance letter]

24 Jul 2023

PONE-D-22-33598R1 

Hydroxychloroquine reduces T cells activation recall antigen responses 

Dear Dr. Fowke:

I'm pleased to inform you that your manuscript has been deemed suitable for publication in PLOS ONE. Congratulations! Your manuscript is now with our production department. 

Kind regards, 

on behalf of

Dr. Janin Nouhin 

Academic Editor

PLOS ONE